# Multi-Resolution Sensing for Real-Time Control with Vision-Language Models

**Saumya Saxena** [*][†][1], **Mohit Sharma**[†][1], **Oliver Kroemer**[1]
Robotics Institute, Carnegie Mellon University[1]
{saumyas, mohits1, okroemer}@cmu.edu

**Abstract:** Leveraging sensing modalities across diverse spatial and temporal resolutions can improve performance of robotic manipulation tasks. Multi-spatial resolution sensing provides hierarchical information captured at different spatial scales and enables both coarse and precise motions. Simultaneously multi-temporal resolution sensing enables the agent to exhibit high reactivity and real-time control. In this work, we propose a framework for learning generalizable language-conditioned multi-task policies that utilize sensing at different spatial and temporal resolutions using networks of varying capacities to effectively perform real time control of precise and reactive tasks. We leverage off-the-shelf pretrained vision-language models to operate on low-frequency global features along with small non-pretrained models to adapt to high frequency local feedback. Through extensive experiments in 3 domains (coarse, precise and dynamic manipulation tasks), we show that our approach significantly improves ($2\times$ on average) over recent multi-task baselines. Further, our approach generalizes well to visual and geometric variations in target objects and to varying interaction forces.

## 1 Introduction

Performing robotic manipulation tasks in the real world often requires using sensing modalities at different *spatial resolutions*. For instance, for peg-insertion, the robot can use a statically-mounted third-person camera (low spatial resolution or global information) to reach close to the hole, use a wrist-mounted first-person camera for finer alignment, and finally use proprioception and force-feedback for insertion (high spatial resolution or local information). Additionally, each sensing modality can be utilized at a different *temporal resolution*. For example, for coarse quasi-static subtasks ("reach hole"), using third-person camera images at a *low frequency* can be sufficient. However, finer reactive subtasks ("insert peg"), might require *high-frequency* force-torque feedback. Based on this insight, we propose a multi-resolution (spatial and temporal resolution) sensor fusion approach for coarse quasi-static as well as precise reactive manipulation tasks.

Multi-resolution sensor fusion can enable generalization to novel visual-semantic targets. For instance, by utilizing global information from third-person camera images only for coarse localization and relying on local information from in-hand cameras and force-torque feedback for finer motions, the policy can learn to generalize to novel objects. Previous approaches to learning generalizable policies either require extensive data collection [1, 2, 3] or rely on pretrained models [4, 5, 6, 7] for policy adaptation [8]. However, such approaches typically utilize a single sensory modality, while others that incorporate multiple sensors do not prioritize generalization [9]. In our work, we avoid extensive data collection and instead leverage pretrained vision-language models in our multi-resolution approach to learning generalizable language-conditioned multi-task policies.

Although pretrained vision or vision-language models (VLMs) provide impressive generalization capabilities and enable learning language-conditioned multi-task policies, using large VLMs can have certain disadvantages. First, given their large size (e.g. Flamingo has 80B parameters [6]), they

---

[*]† equal contribution.

7th Conference on Robot Learning (CoRL 2023), Atlanta, USA.

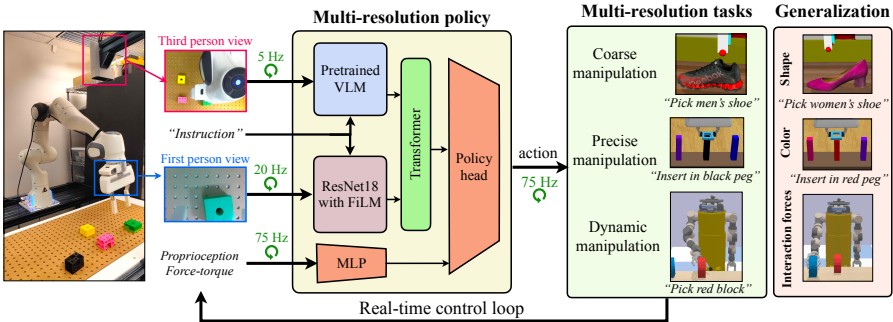

Figure 1: Our proposed approach uses sensing at different spatial and temporal resolutions for real time control of coarse, precise and dynamic tasks while enabling generalization to novel visual features and interactions.

have slow inference which makes them unusable for real-time closed-loop control which is necessary for reactive tasks. Second, since pre-trained models are often trained on out-of-domain data, using them to solve in-domain manipulation tasks (especially precise tasks) may require finetuning [10]. However, task-specific finetuning can make models less robust with reduced generalization [11].

To overcome the above challenges of utilizing large pretrained VLMs for real-time control of reactive tasks, we propose a framework that incorporates different capacity networks (that operate on different sensing modalities) at different frequencies. Specifically, we use large pretrained VLMs with slow inference at a lower frequency while small networks with fast inference at a higher frequency. Our low-frequency pretrained VLMs operate on statically mounted third-person views and can provide global coarse feedback (such as approximate object locations) that is usually only needed at a low rate. On the other hand, we propose using small trained-from-scratch models with first-person camera views and force-torque data to obtain the high-frequency fine-grained feedback necessary to perform precise and reactive tasks. Further, to overcome the challenge of loss in generalization when finetuning pre-trained VLMs, we *freeze* the pretrained VLMs to avoid losing their robustness and maintain their generalization abilities. Overall main contributions include:

- a framework for learning generalizable multi-task policies that incorporates multiple sensory modalities to capture global to local spatial information,
- combine sensor modalities at different frequencies to avoid bottlenecks and enable reactive control which we show empirically is essential for dynamic tasks,
- comprehensive experiments across 3 domains (and 2 real-world tasks) that include coarse, precise and dynamic manipulations tasks, and
- effective generalization across semantic task variations in both simulation and real-world.

## 2    Related work

**Vision-Language Pretrained Models for Robot Manipulation**: Many prior works combine vision and language for robotic tasks. While early works focus on tabula-rasa learning [12, 13, 14], more recent works, use pretrained large language models (LLMs) and show efficient learning and improved generalization for robotics tasks [15, 16, 17, 18, 19]. Many recent works also combine large general-purpose pretrained vision or vision-language models (VLMs) [4, 6, 20] for manipulation [21, 22, 8, 10, 23, 24, 25, 26, 27]. Our work is more closely related to these latter works in that we also use pretrained VLMs for robot manipulation. Among these works, many works only use language for task-specification and do not focus on the generalization provided by pretrained models [26, 27]. Additionally, other works adapt the pretrained representation for the downstream task [24, 10, 28]. However, as we show empirically, such updates lead to representation drift and a loss of robustness for the pretrained general-purpose VLM. Hence, we propose not updating the pretrained representations. While [25, 8] use frozen VLMs, [25] only uses pretrained VLM as an open-world object detector to get pixel targets for the task at the first episode step. On the other hand, [8] uses

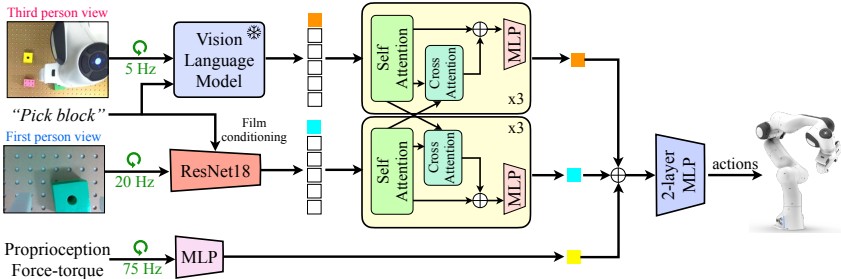

Figure 2: Overall architecture: Global low frequency information is extracted from third-person camera images using slow inference networks, local high frequency information is extracted from first-person camera images and proprioceptive, force-torque feedback using fast inference networks. These sensing modalities are then fused at different frequencies to enable real time high frequency control.

the pretrained VLM with templated pick-and-place actions for manipulation. By contrast, we use VLMs in our multi-resolution framework with continuous feedback for reactive manipulation tasks.

**Multi-Spatial Resolution for Robot Manipulation:** Many prior works use multiple sensor modalities for robot manipulation, wherein each modality operates at a different spatial resolution. For instance, prior works often combine visual (low spatial resolution) and proprioceptive (high spatial resolution) feedback [29, 30, 31], use wrist-mounted cameras for visual servoing [32, 33, 34] or for contact-rich manipulation tasks [35, 36, 37, 38], while other works focus on combining vision and haptic sensing [39, 40, 41, 42]. Our work is similar to the first set of works i.e. we use both third person and first person cameras for precise manipulation. However, unlike most prior works [35, 38] which focus on single-task settings, we focus on multi-task settings and fuse multiple sensing modalities at different resolutions.

**Multi-Temporal Resolution for Robot Manipulation:** Learning reactive policies requires the robot to operate at high frequencies. Some recent works in robot manipulation focus on learning policies at different temporal resolutions. For instance, [43] decompose a manipulation task into different phases (e.g. visual reaching phase and tactile interaction phase) and learn separate policies for each phase as well as a blending policy. While [44] avoid the discrete formulation of an MDP and instead learn a continuous differential equation [45, 46] to model the low resolution features. By contrast, we use the discrete formulation and instead of decomposing policies into different phases we reuse features from low-resolution signals while operating at a high temporal resolution.

**Dynamic Reactive Manipulation:** Many prior works in robot manipulation focus on quasi-static tasks [17, 1]. However, there has been increased interest in solving tasks that are reactive and dynamic in nature [47, 48, 49]. Previous works focus on explicitly learning the dynamics [49] or using analytical models [47, 50] of such systems for achieving reactivity. These works often assume access to the ground truth object pose and are limited to a single-task setting. In our work, we learn how to perform such dynamic and reactive tasks using visual inputs in a multi-task setting.

## 3  Proposed Approach

In this section, we discuss our approach for learning a generalizable language-conditioned multi-resolution multi-task policy for precise and reactive manipulation tasks. Below, we provide details on how we utilize different sensing modalities and then delineate our training/inference and discuss how our approach enables real time control for reactive tasks while generalizing to novel tasks.

### 3.1  Multi-Resolution Architecture

Figure 2 shows the architecture of our multi-resolution approach. Our model takes as input multiple sensing modalities with different spatial resolutions, i.e., statically-mounted third-person camera view, first-person camera view and high frequency force-torque feedback. Each input is first processed separately before being fused together at different temporal resolutions to output high frequency robot actions. Below we expand on each component of our architecture.

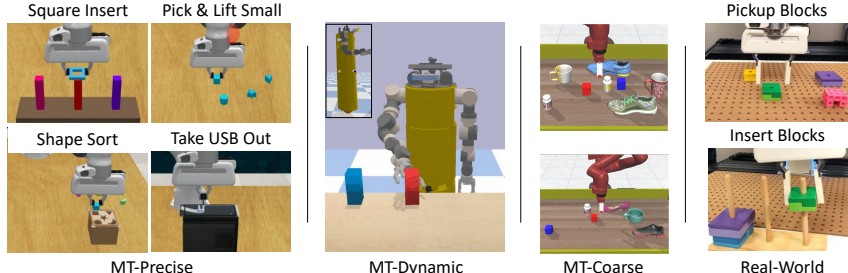

Figure 3: Task settings for evaluating our proposed approach. *Left*: Precision tasks. *Middle-left*: Dynamic tasks. *Middle-right*: Coarse tasks. *Right*: Real world pick and insertion tasks.

**Low-Spatial Resolution Model:** We use a low-spatial resolution sensor (third-person camera) to provide global task information to our agent. We use pretrained visual-language models to extract this global information from third-person views as well as to enable language-conditioning in a multi-task setting. Such pretrained models enable generalization to novel semantic features such as new objects or novel language commands. However, to ensure the pretrained model maintains its robustness we keep it *frozen*. However, using large VLMs to extract this generalizable global information comes with the drawback that the inference speed is very slow ($\approx 5$Hz). We experiment with two models CLIP [4] and MDETR [51] (language-conditioned DETR [52]), which use image-level and object-level information respectively.

**High-Spatial Resolution Model:** To ensure reactivity in the face of slow inference of pretrained VLMs, we use a smaller non-pretrained vision model (ResNet-18) [53] to process the first-person camera view at a higher frequency ($\approx 20$Hz). This view provides us with high-resolution local spatial information. To provide appropriate task-context to the first-person view we use small FiLM layers [54] for language conditioning. We train this model from scratch with augmentations (explained in the next paragraphs) to extract local spatial features that are useful for precise tasks. While using a small vision model enables faster processing it can still be insufficient for some highly dynamic tasks. Hence, we process the force-torque feedback and proprioceptive information at a much higher frequency ($\approx 75$Hz) using a small linear layer.

**Multi-Resolution Sensor Fusion:** We combine local and global sensing information (spatial resolutions) mentioned above at different temporal resolutions based on the capacities of the respective networks. Specifically, we reuse features (network activations) from lower frequency (third-person and first-person views) networks to match the frequency of the highest frequency (force-torque feedback) network. Doing this ensures that the policy network outputs actions at a high frequency (equal to the frequency of the force-torque feedback network), thus enabling real-time control.

In addition to temporal-sensor fusion we also spatially fuse local and global sensing information, i.e, we fuse information extracted from third-person views with first-person view information and vice-versa. We achieve this using two small camera-specific transformers together with cross-attention. Each transformer uses self-attention within each modality (for its associated camera view) and cross-attention with the other modality (other camera view). As shown in Figure 2, we readout the CLS token from each transformer and concatenate them with the force-torque and proprioception embedding. This concatenated embedding is then processed using a 2-layer MLP policy head to output the robot actions. Please refer to the Appendix B for further details on the architecture.

**Data Augmentations:** Data augmentations have been shown to be helpful for single-task learning of manipulation tasks [55, 38]. However, naively using image augmentations can be detrimental for learning generalizable multi-task policies. This is because pixel-level augmentations, such as color-jitter, grayscale etc., can result in semantic changes in the overall scene. Such semantic changes can lead to mismatch between the input image and language instruction provided for the given task. For instance, a demonstration shows "move to red block" but pixel augmentations can change the red block's color. To avoid this while being able to utilize the benefits of augmentations we propose to use two different sets of augmentations. First, for third-person cameras we *only* use image-level augmentations (e.g. random crops, shifts). This avoids mismatch between image-and-text instructions

and allows visual-language grounding from pretrained VLM to be utilized. Second, for first-person camera we use both image-level and pixel-level augmentations (color-jitter, grayscale). Since these augmentations lead to image-text mismatch this further enforces our agent to use the third-person camera view for coarse localization, while only relying on the in-hand view for finer precise motions. Using strong pixel-level augmentations on first-person view further make the in-hand model invariant to texture but rely more on edges and corners [56]. This, as we show empirically, improves the generalization performance of our model on heldout object variations.

**Training and Inference:** We use behavior cloning from expert demonstrations to train our model. We record data from each sensor at their respective frequencies. Specifically, camera images are recorded at 30 Hz and force-torque feedback at 250Hz. To match slower processing times of larger models during inference we sub-sample the third-person camera images to 5Hz and first-person camera images to 20Hz. We use AdamW [57] optimizer with learning rate $1 \times e^{-4}$ and weight decay 0.01. We train our model for 60 epochs, using a linear warmup, starting with learning rate 0, for 5 epochs and then decay the learning rate using a cosine-scheduler. We use a GTX-1080Ti for inference. Overall our architecture has $\approx 250M$ parameters. The pretrained vision-language model has $\approx 150M$ parameters (for MDETR) with an inference time of $\approx 0.1$ seconds. The first-person camera model has $\approx 25M$ parameters with an inference time of $0.04$ seconds. Finally, the force-torque and proprioception model along with the policy head have a total of $\approx 250K$ parameters with an inference time of $\approx 0.005$ seconds. This allows the actions to be inferred at a max frequency of $\approx 200$Hz although we use it at a reduced frequency of $\approx 75$Hz which was sufficient for our tasks.

## 4   Experimental Setup

We first identify the key research questions that we aim to evaluate:

**Q1:** How does **multi-spatial resolution** sensing benefit learning language-conditioned multi-task (MT) manipulation polices for precise tasks? Specifically, we aim to evaluate the utility of multi-spatial resolution sensing for tasks that involve visual occlusions, partial observability, and precision.

**Q2:** How does **multi-temporal resolution** sensor fusion benefit learning reactive manipulation tasks? Specifically, we evaluate how our architecture enables closed loop control for reactive tasks.

**Q3:** How well does our approach **generalize** to tasks with novel visual-semantic targets? Specifically, we evaluate our approach's robustness to distribution shifts, e.g., object colors and geometries.

### 4.1   Environments

To evaluate the above questions we use three task settings, 1) **MT-Precise**: Precise manipulation tasks, 2) **MT-Dynamic:** Dynamic manipulation tasks, and 3) **MT-Coarse:** Coarse table-top manipulation tasks. Below we detail each environment and discuss its usage to answer above questions.

**MT-Precise**  For precise manipulation we use 4 spatial precision tasks from RLBench [58] (see Figure 3 (Left)) – square block insertion, pick up small objects, shape sorting, and unplug usb. We use this task domain to answer **Q1**. Specifically, we evaluate the need for multi-spatial resolution sensing in manipulation tasks that require precise feedback and have partial observability, i.e., objects can go out of view of the first-person camera.

**MT-Dynamic:** We use the CMU ballbot [59] platform to perform dynamic pickup tasks in simulation (Figure 3 (Middle-Right)). We choose ballbot since it is a highly dynamic robot with an omnidirectional base (ball) capable of performing fast, reactive and interactive tasks. We consider the task of dynamically picking up an object, which requires quick reaction to contact with the object and grasping it to prevent toppling the object over. We use this setting to answer **Q2**.

**MT-Coarse:** We consider a canonical table-top manipulation setting ([60, 61]) involving coarse pick-and-place manipulation tasks with diverse objects – blocks, shoes, mugs, cylinders. We use this environment to answer **Q1** and **Q3**. Specifically, for **Q1** we contrast these coarse manipulation tasks with high precision tasks to evaluate the utility of multi-spatial resolution sensing.

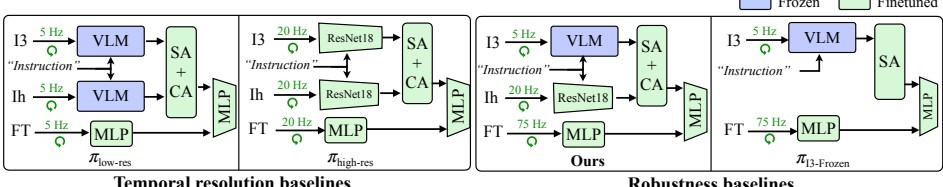

Figure 4: Temporal resolution and robustness baselines used to compare our multi-resolution approach.

**Real-World Setup:** We evaluate our approach on two real-world tasks. For precise manipulation (**Q1**) we use an *insertion* task to insert different blocks into cylindrical pegs (Figure 3 (Right top)). We also evaluate generalization abilities (**Q3**) using a *pickup* task, wherein we use 2 train objects and evaluate the learned policy on 8 objects with different geometry (shape, size) and visual (color, texture) features. Additional details on each environment are provided in Appendix A

### 4.2 Baselines

We compare our approach against recent methods which focus on learning generalizable policies in multi-task settings. We compare against RT-1[1] which proposes a transformer based policy and also against BC-Zero [2] which uses language conditioning using FiLM [54]. However, both [1, 2] focus on coarse manipulation tasks and operate at a single-resolution (both temporal and spatial). To the best of our knowledge no prior work focuses on a multi-resolution approach for multi-task learning. Hence, to highlight the benefit of each component of our approach and answer the questions posed in Section 4 we modify our approach along different axes and propose additional baselines below.

**Spatial Resolution baselines:** To verify the utility of multiple spatial resolutions (**Q1**) we modify our approach and *remove* one sensory modality at a time. We use $\pi_{-Ih}$, $\pi_{-I3}$, $\pi_{-FT}$ to refer to policies which *remove* first-person (**h**and view), **3**rd person view and **f**orce-**t**orque respectively.

**Temporal Resolution baselines:** To answer **Q2** we compare against single temporal-resolution approaches (Figure 4 (Left)), i.e., where all modalities (including force-torque) operate at the same frequency. We introduce two baselines, 1) $\pi_{high-res}$ : small models with fast inference for both cameras (20Hz), and 2) $\pi_{low-res}$ : larger models with slow inference for both cameras (5Hz).

**Robustness baselines:** We compare visual-semantic generalization ability of our approach (**Q3**) against two baselines (Figure 4 (Right)): 1) $\pi_{multi-res-FT}$: Finetune the pretrained VLM model, 2a) $\pi_{I3-Frozen}$: Uses only third-person camera (and force-torque) and keeps the pretrained model frozen. 2b) $\pi_{I3-FT}$: Uses only third-person camera (and force-torque) but finetunes the pretrained model.

**Metrics:** We use task success as the evaluation metric and report mean success over all tasks. During training, we evaluate the policy every 4 epochs and report average over *top-5* mean success rates across all evaluation epochs. For task generalization (**Q3**) we evaluate the train policy on novel visual-semantic tasks not seen during training. For all evaluations we use 20 rollouts per task. Further training details are provided in Appendix B.1.

## 5 Experimental Results

First, we evaluate the effectiveness of our multi-resolution approach against common multi-task baselines, RT-1[1] and BC-Zero[2]. We then present results for each research question. For qualitative results see: https://sites.google.com/view/multi-res-real-time-control.

### 5.1 Comparison to Multi-Task Baselines

Table 1 shows the results for the multi-task baselines RT-1[1] and BC-Zero[2] across all task We note that for coarse manipulation tasks (MT-Coarse) these baselines, that use single camera views, can perform quite well. This is because these tasks only require coarse localization of the target object for task completion. However, for precise manipulation tasks (MT-Precise), such baselines perform quite poorly since these tasks require fine-grained grasping (as many objects are $\approx$ 1cm in size) and insertion for successful task completion.

|  | $\pi_{-Ih}$ | $\pi_{-I3}$ | $\pi_{-FT}$ | Ours |
|---|---|---|---|---|
| MT-Coarse | 74.5 | 41.0 | 81.8 | 82.0 |
| MT-Precise | 7.7 | 29.6 | 56.1 | 55.0 |
| MT-Dynamic | 65.8 | 27.5 | 33.2 | 73.6 |

Table 2: Results for multi-spatial resolution experiments (Section 5.2). Here, − implies that we remove this input from policy. Thus, $\pi_{-Ih}$ implies that the policy only operates on third-person camera views and force-torque feedback.

|  | $\pi_{\text{low-res}}$ | $\pi_{\text{high-res}}$ | Ours |
|---|---|---|---|
| MT-Coarse | 82.0 | 81.0 | 82.0 |
| MT-Precise | 53.4 | 56.2 | 55.0 |
| MT-Dynamic | 4.2 | 12.2 | 73.6 |

Table 3: Results for multi-temporal resolution experiments (Section 5.2). Here, both $\pi_{\text{low-res}}$ and $\pi_{\text{high-res}}$ are single-resolution approaches which run at 5 Hz and 20 Hz respectively, while ours is a multi-resolution approach.

|  | $\pi_{\text{I3-Frozen}}$ | $\pi_{\text{I3-FT}}$ | $\pi_{\text{multi-res-FT}}$ | Ours |
|---|---|---|---|---|
| MT-Coarse (Visual) | 74.5 / 7.1 | 81.8 / 25.8 | 82.4 / 45.6 | 82.0 / 72.3 |
| MT-Coarse (Geometry) | 44.2 / 16.8 | 56.4 / 18.4 | 60.7 / 31.9 | 58.9 / 44.6 |
| MT-Precise (Visual) | 7.7 / 4.5 | 15.6 / 9.2 | 56.4 / 31.9 | 55.0 / 48.1 |

Table 4: Robustness experiment results, each cell shows *train/heldout* success rate (Section 5.2)

domains. On the other hand, our multi-resolution approach, performs much better as it uses the first-person camera view and force-feedback for finer grasping and insertion. For dynamic tasks (MT-Dynamic), our method considerably outperforms the baselines (1.5x). This is because dynamic tasks require *reactive* response to contact events. Only our multi-temporal resolution approach utilizes high spatial and temporal resolution sensing, enabling fast response to contact events.

|  | MT-Coarse | MT-Precise | MT-Dynamic |
|---|---|---|---|
| RT-1 | 81.0 | 12.5 | 4.5 |
| BC-Z | 74.1 | 7.8 | 4.8 |
| Ours | 82.0 | 55.0 | 73.6 |

Table 1: Task success comparison for multi-task baselines across all task domains.

## 5.2 Additional Baseline Comparisons

**Q1 – Spatial Resolution Experiments:** We now compare against the *spatial* resolution baselines discussed in Section 4.2. For this set of baselines all methods use multi-*temporal* resolution sensing with high-frequency force-torque feedback. Table 2 shows results across all task settings. For MT-Coarse we see that only using a first-person camera ($\pi_{-I3}$) performs poorly. This is because of partial observability in this view, i.e., the target object can be out of view and lead to task failure. On the other hand, for MT-Precise (Row 2), only using first-person camera ($\pi_{-I3}$) performs better ($\approx$ 2×) than using only the third-person camera ($\pi_{-Ih}$). This is because MT-Precise tasks require finer motions which are hard to perform from low spatial resolution (third-person) view only. Further, for dynamic tasks (Row 3), using first-person views alone again suffers because of partial observability.

**Q2 – Temporal Resolution Experiments:** Table 3 compares against single-temporal resolution baselines ($\pi_{\text{low-res}}$ and $\pi_{\text{high-res}}$ ). Table 2 shows that for coarse and precise domains single-resolution perform as well as our multi-resolution approach. This is because tasks in both domains are quasi-static and hence fast reaction to contact events is not critical for task success. On the other hand, for dynamic tasks (Table 2 bottom row), since fast response to contact events is necessary (to avoid failures such as object toppling, see Figure 7 Appendix) our multi-resolution approach performs better than both $\pi_{\text{low-res}}$ (5Hz) and $\pi_{\text{high-res}}$ (20Hz) since it incorporates force feedback at 75Hz.

**Q3 – Robustness Experiments:** Table 4 compares results *(train / heldout)* for visual-semantic generalization against the robustness baselines in Section 4.2. As noted previously, for these experiments we evaluate the trained policies on *heldout* environments (see Appendix B.1 for details). We note that our approach, with frozen pretrained model, generalizes better than the finetuned model $\pi_{\text{multi-res-FT}}$. This shows the ability of our approach to maintain the generalization capabilities of the pretrained VLM as compared to the finetuned model that suffers from 'forgetting' and representation drift towards the training tasks. Additionally, from column-1 and column-2, we again note that the finetuned $\pi_{\text{I3-FT}}$ model suffers a larger decrease in performance as compared to $\pi_{\text{I3-Frozen}}$. Finally, comparing $\pi_{\text{I3-FT}}$ against $\pi_{\text{multi-res-FT}}$, we see that even with finetuning our multi-spatial resolution approach generalizes better because it can utilize first-person views for improved task success.

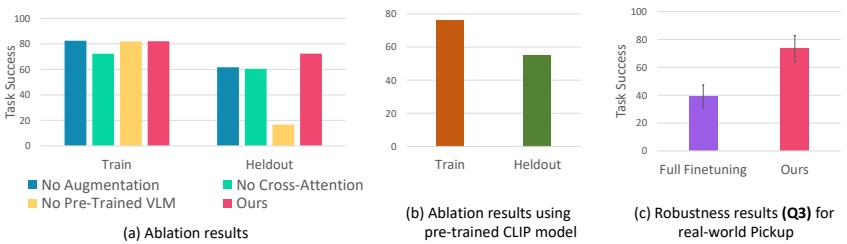

(a) Ablation results

(b) Ablation results using pre-trained CLIP model

(c) Robustness results **(Q3)** for real-world Pickup

Figure 5: *Left:* Ablation results (see Section 5.3). *Right:* Robustness result for real-world pickup.

**Real-World Experiments:** We evaluate our approach in the real-world on two tasks, pickup and peg-insertion [62]. Table 5 shows comparison against the spatial resolution baselines. We note that our approach, with multi-spatial resolution, performs $\approx 3\times$ better than the baselines on both tasks. We see that given *limited* demonstrations both $\pi_{-I3}$ and $\pi_{-Ih}$ fail to perform well (across both tasks). On the other hand, removing force-torque feedback $\pi_{-Ih}$ only affects performance on insertion task ($\approx 20\%$ less) since this task relies more on contact feedback. Additionally, Figure 5 (c) figure plots the robustness result for pickup task. As before we see that our approach with frozen model performs better. See website for qualitative results.

|  | $\pi_{-Ih}$ | $\pi_{-I3}$ | $\pi_{-FT}$ | Ours |
|---|---|---|---|---|
| Pickup | 7.5 (3.5) | 20.0 (14.1) | 67.5 (3.5) | 75.0 (7.0) |
| Peg-Insert | 10.0 (0.0) | 12.5 (4.6) | 42.5 (3.5) | 67.5 (3.5) |

Table 5: Mean (stdev) results (using 2 seeds) for multi-spatial resolution for real world tasks.

## 5.3 Ablations

We further ablate the different components of our proposed approach. Due to space limitations we only summarize key findings and provide details in Appendix C.2.

**Pixel-Level Augmentations:** For pixel-level augmentations (Figure 5 (a) blue bar) we see little difference in training performance but larger increase in generalization performance $\approx 15\%$.

**Spatial Sensor Fusion using Cross-Attention:** Figure 5 (a) (green bar) shows that using concatenation instead of cross-attention reduces performance ($\approx 10\%$) on both train & heldout tasks.

**Effect of Pretraining:** We also evaluate the effects of using pretrained-VLMs. Figure 5 (a) (yellow bar) shows that not using a pretrained model (no vision-language grounding) suffers little drop in train performance but significant drop ($3\times$ worse) in generalization, i.e. heldout performance.

## 6 Conclusion and Limitations

Our work proposes using sensing modalities at multiple spatial and temporal resolutions for learning multi-task manipulation policies. Our multi-resolution approach captures information at multiple hierarchies and allows the robot to perform both coarse and fine motions with high reactivity and real-time control. To learn generalizable multi-task policies we further leverage off-the-shelf pretrained vision-language models and freeze them to maintain their robustness. Our work has several limitations. While our proposed framework is general for multi-spatial sensing we only rely on global third-person camera and local first-person camera view. Further local sensing using vibro-tactile sensors [63, 64, 65] was not explored. Further, it is unclear if our approach of using cross-attention for sensor fusion will be optimal for more than 2 sensors. Additionally, while our multi-resolution policy allows us to learn robust policies not all sensing modalities will be available for all tasks. Thus, future work should explore adapting to scenarios with missing sensing modalities.

## Acknowledgements

This project was supported by NSF Grants No. CMMI-1925130 and IIS-1956163 ONR Grant No. N00014-18-1-2775, ARL grant W911NF-18-2-0218 as part of the A2I2 program.

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

# A   Environment Details

In this section we provide further details on the different environments used in our experiments.

## A.1   MT-Coarse Manipulation

For coarse manipulation tasks we focus on a variety of objects including blocks, mugs, cups, and shoes (both men and women shoes). As noted in the main paper, for these set of objects we focus on pick-and-place skills. However, we note that we did experiment with more complex contact-rich skills (e.g. pushing, stacking). However, we found the physics to be unstable with more complex objects (e.g. cups). For instance, pushing cups would almost always topple them and roll over. For future work, we hope to make our skills more robust.

Specifically, we use fixed size blocks with different semantic colors, 4 mugs, 4 cups and 4 shoes. We use google scanned objects [66] to collect non-block objects and use mujoco [67] to simulate our environment. We use the latest mujoco environments to import meshes into the simulator. Each environment in this set of tasks is created by first selecting a target object-type and then selecting a target object from the set of objects. We then select 3-5 distractor objects to fill the scene. These objects are uniformly selected from the remaining objects.

## A.2   MT-Precise Manipulation

As noted in the main paper for precise manipulation tasks we use the spatial precision set of tasks from RLBench [58]. Overall, we use 4 tasks (see Figure 3 (Left)) – square block insertion, pick up small objects, shape sorting, and unplug usb from computer. We avoid using the motion-planner augmented approach for solving these tasks and instead opt for learning reactive closed-loop control policies. We use the delta end-effector actions for our tasks. Additionally, we use standard front and wrist mounted camera. along with proprioceptive and force-torque feedback as policy input.

However, directly using end-effector actions increases the policy horizon significantly. Moreover, naively using the original input distribution for each task also requires learning full 6-DOF policies. Both of these can significantly increase the data requirements to learn the manipulation policy. To avoid this we restrict the starting distributions for each task such that the objects are spawned in a slightly narrow region infront of the robot. We further make other task-specific changes, detailed below, such that the robot can perform each task without changing hand orientations.

**Insert Onto Square Peg:** For this task we restrict the orientations of the square ring (blue object) and the peg on which to insert. This allows the robot to perform the task without changing gripper orientations. Further, we use a region of $40cm \times 30cm$ infront of the robot to spawn both the base and ring. Finally, the default task configuration provides 20 different peg colors, of which we use the first 10 colors for training and remaining 10 colors for robustness experiments.

**Pick and Lift Small:** For this task, we again use a region of $40cm \times 30cm$ infront of the robot to spawn both all objects. We also restrict the orientation of each object such that it can be grasped directly without requireing gripper orientation changes.

**Shape-Sorting:** The default configuration for the shape-sorting task considers 4 different shaped objects (see Figure 3 Bottom-Left) – square, cylinder, triangle, star, moon. In the default RLBench configuration most objects directly stick to the robot finger and are simply dropped into the hole for task completion. However, with closed loop control we find that non-symmetric objects (star, triangle, and moon) can have significant post-grasp displacement such that it is impossible to insert these objects without changing gripper orientation. Hence, we exclude these two objects from evaluation and only use symmetric square and cylinder objects.

**Take USB Out:** This task requires the robot to unplug a USB inserted into the computer. However, the default configuration for this task requires 6-dof control. To avoid this, we create smaller computer and USB assets and mount them vertically on the table such that the USB can be unplugged without changing hand orientation. See Figure 3 (Bottom-Right) for visualization.

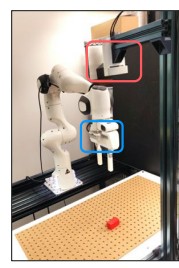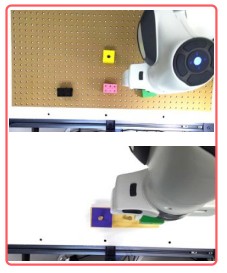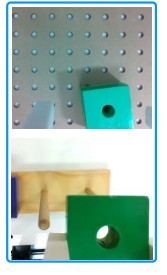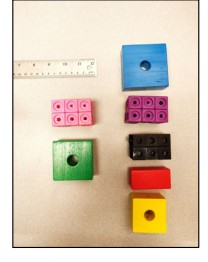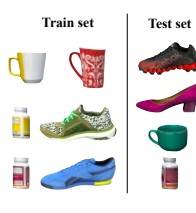

(A) Real-world setup for pickup and insertion tasks.    (B) Examples objects for real-world  pickup task    (C) Example objects for coarse manipulation task

Figure 6: *Left:* Real World env setup with third-person (red) and first-person (blue) camera views. *Middle:* Example objects set used for real-world pickup task. *Right:* Example objects used for MT-coarse.

## A.3   MT-Dynamic Manipulation

This task involves using the CMU Ballbot in simulation (PyBullet [68]) to perform a dynamic pick up task. The task involves picking up a block that is placed on a table in front of the ballbot. We use two blocks (red and blue) in this task and use language instructions to specify which object to pick up. The initial conditions are set such that the table and objects are always out of the reach of the ballbot arms and the ballbot has to roll forward to pick up the objects. We use a statically mounted camera looking at the table and the ballbot as the third-person camera and the camera located on the turret of the ballbot as the first-person camera. The turret tilt is adjusted such that the objects on the table are initially out of the view of the turret camera and only when the ballbot starts moving towards the table, the objects come into view. The third person camera is always able to view both the objects and the ballbot. We use task space control to control the ballbot end-effector while a center of mass balancing controller is always running in a high-frequency feedback loop to balance the ballbot.

## B   Architecture Details

Section 3 discusses the overall architecture used in our work. To recall, our proposed architecture uses a multi-resolution approach with multiple-sensors, each with different fidelity. We process each sensor with a separate network which is conditionally initialized using a pre-trained vision-language model. The output of each vision model is flattened to create a set of patches. For DETR [51, 52] based model we use a ResNet-101 backbone and flatten the output layer into 49 patches and add positional embedding to it. For CLIP [4] we use a ViT-B model and use hierarchical features from the 5'th, 8'th and 11'th layer. Since MDETR already does vision-language fusion using a transformer we directly use its output. However, since CLIP only weakly associates vision and language at the last layer, we additionally use FiLM layers to condition the output. Our use of FiLM is similar to previous models [69]. For each camera modality we use a small transformer with multi-head attention. Each transformer uses an embedding size of 256 and 8 heads. We use post layer-norm in each transformer layer. Further, in each transformer layer we use cross-attention with the other camera. Overall we use 3 transformer layers for each camera modality. Our force-torque and proprioceptive input is concatenated together and mapped into 256 dimensions using a linear layer. We concatenate the readout tokens from each camera transformer and the force-torque embedding. This $256 \times 3$ size embedding is then processed by 2 linear layers of size 512 which output the robot action.

**Input:** For each of our camera sensor we use an image of size $224 \times 224$. For proprioceptive input we use the end-effector position of the arm. While for force-torque input we use the 6 dimensional force-torque data. We use cropping augmentation for both camera sensors. Specifically, we first resize the image to 226 and then do random crop with shift $= 8$. For, more aggressive pixel level

| Key | Value |
| --- | --- |
| batch size | 16 |
| proprio and force torque embedding | 256 |
| camera-transformer embedding Dim. | 256 |
| camera-transformer feedForward Dim. | 768 |
| Number of transformer layers | 3 |
| learning rate | 0.0001 |
| warmup epochs | 5 |
| total epochs | 60 |
| optimizer | AdamW |
| weight decay | 0.01 |
| scheduler | cosine |

Table 6: Hyperparameters used for our architecture and model training.

augmentations we stochastically apply grayscale and use color jitter with brightness $\in (0.4, 0.8)$, contrast $\in (0.4, 0.8)$, saturation $\in (0.4, 0.6)$ and hue $\in (0.0, 0.5)$. These augmentations significantly change the underlying visual semantics of the task.

## B.1 Training Details

In this section we provide details on the demonstrations (for each environment type) used to train our approach. Further, we also provide details on the train and heldout configurations used for robustness evaluation.

**MT-Coarse:** As noted above in Appendix A.1, we use multiple different objects to train and evaluate our policy. Each environment is created by first sampling a target object and then a set of distractor objects. For each environment and skill combination we collect 20 demonstrations. Overall, this gives us $\approx 1000$ demonstrations across all tasks. We then learn one policy across all tasks.

**MT-Precise:** For spatial precision tasks from tasks from RLBench [58] we use 4 different tasks. As discussed in Section A.2, each task has it's own set of variations. For training our multi-task policy we use try to balance the number of demonstrations from each task. For square peg insertion (*insert onto square peg*) task we use first 10 variations for training and gather 25 trajectories per variation. Each other task has less than 4 variations hence for each task we use 100 demonstrations each for training. To test visual-semantic robustness for these tasks Section 5.2 we use the insert-onto-square-peg task since only this task has any semantic variations. We use the remaining 10 peg colors (i.e. 10 heldout variations) to test each approach.

**MT-Dynamic:** To collect expert demonstrations, we sample the locations of the objects on the table in a 70cm*20cm region and sample the initial ballbot location in a 50cm*50cm region. We collect 50 demonstrations for each task setting (each block). As noted earlier, the third-person camera is used at a frequency of 5Hz, the turret camera is used at 20Hz and proprioception and force-torque feedback is used at 75Hz.

**Real-World:** For real-world tasks we collect data using teleoperation with a leap-motion device which can track hand movements upto a 100Hz. We map these movements to robot movements and collect proprioceptive and force-torque data at 75Hz, while both cameras are recorded at 30Hz. To collect data for pickup tasks we use two blocks with different shapes and different colors. The green and pink blocks in Figure 6 (Right) were used to collect all training data. While evaluation happened on 8 other blocks, each with a different shape and color. For training our policies we collect 60 demonstrations for each pickup variation and 50 demos for the insertion task. We note that the initial state distribution for insertion was narrower than pickup and hence it required fewer demonstrations.

**Metrics:** We use task success as the evaluation metric. Since we use a multi-task setting we report mean success over all tasks. During training, we evaluate the policy every 4 epochs on all train tasks. We report the average over *top-5* mean success rates across all evaluation epochs. For task generalization results (**Q3**) we use the trained policy and evaluate it on novel visual-semantic tasks

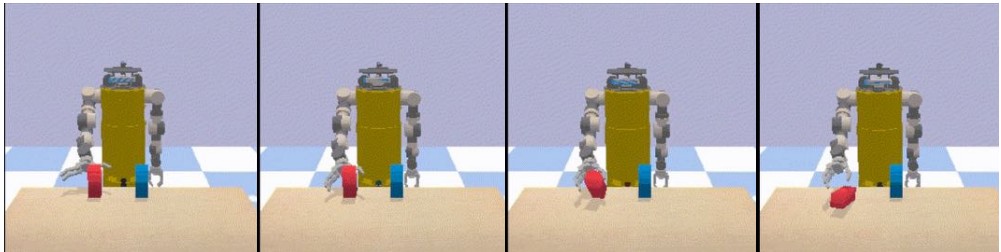

Figure 7: Example failure case for MT-Dynamic (Ballbot) task. As can be seen in the figure, if the robot approaches the object but does not react fast enough to the object contact, the block can topple resulting, in task failure.

which were never seen during training. Hence, for Q3 we report task success on novel unseen tasks. For all evaluations we use 20 rollouts per task. Further training details are provided in Appendix B.1.

## B.2 Implementation Details

In this section, we discuss our real-robot implementation details. In our implementation, the real-time control loop is composed of a low-level task-space impedance controller and a high-level neural network controller. The low-level controller operates at 1KHz using a real-time kernel and sends control commands to Franka Panda's control interface (FCI) [62]. Our neural-network controller implementation can operate up to a maximum of 100Hz given communication latency. Specifically, for our experiments we run the neural network controller at 75 Hz. We use fixed low impedance values (Kp: 350) to avoid damaging the robot during fast execution of contact-rich tasks.

**Neural network controller implementation:** For our real-robot neural-network controller implementation we follow a multi-threaded architecture. Robot state information such as proprioceptive data and force-torque data is published at 100Hz, while camera images are published at 30Hz. Each sensor modality is appended to a separate fixed size time-stamped buffer. We process each modality independently in a multi-threaded manner by extracting the latest time-stamped data from the respective buffer.

Camera images are processed on separate threads using their respective neural networks and we save the network outputs for future processing. More specifically, we process images from third-person camera using a large VLM and save a set of visual-language representations from its output in a buffer. This thread is limited by the inference speed of the large VLMs and operates at 5Hz. We process the image from the in-hand camera in a separate thread using a small ResNet based model to get hand-camera image representations. On the same thread, we further process these hand-camera image representations with the existing cached vision-language representations using cross-attention layers to get multi-modal fused visual-language output which is added to a fixed size buffer. This thread operates at 20Hz.

Finally, the high-level neural network controller (which runs on the main thread at 75Hz) concatenates the cached robot state information (force-torque, proprioceptive) with the latest fused multi-modal features. The concatenated features are processed through a small multi-layer perceptron to get the final action output which is sent to the low-level impedance controller.

## C Additional Results

### C.1 Additional Real-World Comparisons

In addition to real-world results in Table 5 we also tried out BC-Z and RT-1 on the pickup task in the real world. Table 7 reports the average success rate and compares them to our method. We find that BC-Z's performance is much worse than our proposed approach. This is because BC-Z operates at a single-resolution (both spatial and temporal) as it uses only a third-person camera. In the absence

| Setup | BC-Z [2] | RT-1 [1] | Ours |
|-------|----------|----------|------|
| Train | 12.5 | 0.0 | 75.0 |
| Eval | 5.0 | 0.0 | 71.1 |

Table 7: Real-World results for using commonly used imitation learning (single-spatial resolution baselines) for Pickup task.

| | $\pi_{\text{low-res}}$ | $\pi_{\text{high-res}}$ | Ours |
|---|---|---|---|
| RealWorld - PegInsert | 45.0 | 62.5 | 67.5 |

Table 8: Additional Results for multi-temporal resolution experiments. As before, both $\pi_{\text{low-res}}$ and $\pi_{\text{high-res}}$ are single-resolution approaches which run at 5 Hz and 20 Hz respectively, while ours is a multi-resolution approach.

of a first-person camera view it is often unable to accurately localize the target object and fails to perform the final fine-grained motion to grasp the object and lift it up. Further, for RT-1 we find the performance to be very poor. We believe this is because RT-1 uses tokenized actions which requires us to discretize our continuous robot actions. Since we operate in the low data regime (120 trajectories) such discretization leads to token imbalances during training and deteriorates the model's performance. Additionally, since RT-1, similar to BC-Z, uses single-resolution (i.e. third-person camera only) we believe its performance suffers from similar challenges of inaccurate localization. Furthermore, we evaluate visual generalization of the BC-Z and RT-1 policy on novel unseen objects (and instructions). Since both BC-Z and RT-1 do not use a pre-trained vision-language model and thus have no visual grounding for the text instructions they fail to perform well on unseen novel objects. By contrast, our approach that utilizes a pretrained VLM generalizes well.

## C.2 Additional Ablations

We further ablation on the different components of our proposed approach. For these set of results instead of using all 3 environment suites for evaluation, we choose the most appropriate environment suite for each component of our approach and evaluate on it.

**Pixel-Level Augmentations:** We evaluate the effect of pixel-level augmentations (color jitter, grayscale) on the training and generalization of our MT-policies on MT-Coarse. Figure 5 reports results on both training and heldout (novel) evaluation configurations. We see that while there is very little difference in training performance, extensive pixel-level augmentations helps generalization by close to $\approx 15\%$. While pixel-level augmentations change the semantics of the task, our multimodal approach is still able to complete the task because of visual-language grounded provided from pretraining.

**Multi-Modal Fusion using Cross-Attention:** We compare use of early fusion using cross-attention with late fusion using concatenation. Figure 5 shows that using cross-attention improves the performance by around $\approx 8\%$ on both train and heldout configuration. Thus, using cross-attention for multi-modal fusion is more effective than concatenation. However, we note that cross-attention requires more parameters and has slower inference.

**Effect of Pretrained-VLMs:** We also evaluate the effects of using pretrained-VLMs. Figure 5 shows the training and heldout performance using ImageNet initialization which only has visual pretraining and no vision-language pretraining. We see that while training performance matches our approach the heldout performance decreases tremendously. This large decrease is due to missing visual-language grounding since we use *separately* trained visual and language models.

**Real-World Temporal-Resolution Comparison:** We also ablate the effect of temporal resolutions on real-word robot performance. Specifically, we evaluate single temporal-resolution approaches ($\pi_{\text{low-res}}$) and $\pi_{\text{high-res}}$ for the peg-insertion task in the real-world. As before, to evaluate the learned policy we run each episode for a fixed duration of 60 seconds. However, we use early termination if the episode is solved successfully or the robot violates the desired workspace. Table 8 shows our results. Given that the insertion task is not dynamic, $\pi_{\text{high-res}}$ performs similarly to our approach. However, by comparison, ($\pi_{\text{low-res}}$) performs much more poorly (45% only). This is because a low-temporal resolution policy is not very reactive and hence doesn't respond fast to contacts made with the wooden peg. Thus, it is often unable to find the appropriate location to insert the block into the

wooden peg. This can also be seen from qualitative videos (see success and failure videos), where both success and failure scenarios are much less reactive.

**Temporal-Resolutions:** Finally, we also ablate the temporal frequencies for the MT-Dynamic tasks. We ablate the effect of using camera inputs at low-resolution (third-person and in-hand camera inputs at 5Hz) while only force-torque feedback is used at high-resolution (75Hz).

Table 9 below shows our results. From the table below, we observe that the performance on MT-Dynamic tasks drops significantly when using the camera views at a very low temporal resolution. From our qualitative observations we note two common failure cases. First, where the ballbot is sometimes unable to reach the block to pick up. This is because, due to latency in the camera inputs (5 Hz), the policy outputs sub-optimal actions. Upon receiving updated camera inputs the policy tries to correct the trajectory. The overall resulting trajectory is noisy and fails to reach the target object. Second, again due to camera latency, the end effector does not align well with the target object and ends up toppling the object while trying to grasp it.

| $\pi_{\text{low-res-high-FT}}$ | Ours |
|---|---|
| 33.4 | 73.6 |

Table 9: Results for using low-temporal resolutions for camera-inputs (5Hz) and high-temporal resolutions for force-torque only (75Hz).

