# OpenReview forum: "Multi-Resolution Sensing for Real-Time Control with Vision-Language Models"
_robot-learning.org/CoRL/2023/Conference — CoRL 2023 Poster_

### Official Review · Reviewer_FtkG · 2023-07-17

**Confidence:** 4
**Originality:** Very Good
**Technical Quality:** Good
**Clarity Of Presentation:** Good
**Impact:** 4

**Recommendation:**

Weak Accept: I recommend accepting the paper, but will not argue for my recommendation if the majority of other reviewers have a different opinion.

**Review:**

Strengths:
- (Significance) This work presents a framework for tackling dynamic and precise tasks with pre-trained vision-language models. This allows the method to be sample-efficient and tackle challenging robotics tasks.
- (Originality) This is the first work that I'm aware of that leverages VLMs for dynamic/precise tasks.
- (Quality) There are some results on a real (albeit, relatively simple) robotic task.

Weaknesses:
- (Clarity) I think the messaging could be improved. Currently, there are two axes of information discussed: spatial and temporal. However, it ties together the two, i.e., low-resolution spatial info is to be processed at a low frequency, which is not always the case. For some tasks, global information may need to be processed at a high frequency. What would be clearer is to first focus the problem on designing a general-purpose (static) manipulation policy; then motivate the use of VLMs for this domain; then explain the need for high-frequency processing of finer feedback such as force.

- (Significance) While the simulated results are quite varied, the generalization studied in the real robot setup is considerably narrower.

**Quality Of The Limitations Section:**

Limitations are addressed clearly

**Questions For Rebuttal:**

- How does performance vary with the size of the training dataset?
- Is there a benefit to processing inputs at a higher frequency than studied right now?
- In the real robot setup, the robot partially occludes the workspace in the third-person view. Is a first-person view still needed if the camera is placed so that there is less occlusion?
- Could this approach enable handling of dynamic instructions?


**Robotics Focus:**

Sufficient demonstration on hardware

**Summary Of Paper:**

This paper studies generalization in policies that operate at different spatial and temporal resolutions. The main motivation is that different parts of a task may require different sensors (e.g., third-person camera images for global information versus a wrist-mounted camera for finer feedback) and, may either require high-frequency or low-frequency information processing (i.e., at different temporal resolutions). This work leverages pre-trained vision-language models to process the global information at a lower-frequency to avoid the need for large amounts of data collection.

**Summary Of Recommendation:**

This work contributes a framework that leverages the strengths of pretrained VLMs while mitigating their weaknesses, i.e., slow inference time and inability to handle non-third person view inputs. The experiments show that augmenting them with trained-from-scratch networks to handle high-resolution spatial and temporal information improves performance on dynamic and precise manipulation tasks.

---

### Official Review · Reviewer_VHHQ · 2023-07-20

**Confidence:** 4
**Originality:** Good
**Technical Quality:** Good
**Clarity Of Presentation:** Very Good
**Impact:** 3

**Recommendation:**

Weak Reject: I recommend rejecting the paper, but will not argue for my recommendation if the majority of other reviewers have a different opinion.

**Review:**

*Originality*: To my knowledge, the proposed method is a novel combination of individually well known techniques, but I still believe their analysis to be valuable. The use of hand cameras and third person cameras together is a common choice in the literature.

*Clarity*: The paper is well written and organized, it was simple for me to understand the key contributions. However I do find the paper to be a bit crowded, the temporal and spatial aspects for example are somewhat independent, and I overall think the paper should be separated out and analyzed in more detail for each claim.

*Quality*: The methods and claims are technically sound.

*Strengths*
- The results support the claims made in the paper. More dynamic tasks require F/T information at a higher frequency, and occlusion is partially prevented with the use of both hand and third person views. Pre-trained VLMs offer better generalization than models trained from scratch or fine-tuned.
- The authors provided many interesting baselines and a good range of experiments in simulation.

*Weaknesses*
- I'm curious why the authors did not consider any recurrent architectures for their method? The use of recurrence might affect claims about the need for various temporal and spatial resolutions, and so this would be good to see in the rebuttal.
- It is unclear to my why the tasks in MT-Dynamic are considered "dynamic" given that they are just pick up tasks. The provided justification of potentially knocking over objects in my mind is not a strong enough justification for calling these dynamic. I'd be interested to see more traditionally dynamic tasks, like scooping objects or throwing trash away, or anything that requires complex f/t and end effector signals. In general the provided tasks were somewhat simple.
- I think there should be a bit more experimentation with temporal frequencies of each input modality. Some examples: what happens when 3rd person is run at 20Hz but 1st person is run at 5Hz (testing if there's any benefit to higher frequency hand camera views)? or when both are run at 5Hz but F/T is still run at 75Hz (testing if we can get away with slower visual perception but faster proprioception)? In general, as a consequence of having three separate main ideas in this paper, I felt like there were a lot of experiments but not enough evidence for each claim (e.g., choice of VLM, where to incorporate it, how to fuse sensory feedback and how that changes at different frequencies, etc.)

**Quality Of The Limitations Section:**

Limitations are addressed clearly

**Questions For Rebuttal:**

See above points.

**Robotics Focus:**

Sufficient demonstration on hardware

**Summary Of Paper:**

This paper has three main ideas around imitation learning: (1) incorporating multiple spatial resolutions (in this case, hand and third person views) is better, (2) incorporating multiple frequencies of control for different modalities (high frequency f/t) is better, and (3) VLMs provide more robust low resolution representations for visual and geometric variations. Their model consists of a low-frequency VLM encoder for the third person view fused with medium-frequency Resnet encoding of the hand camera view using a cross attention mechanism. Additionally, force/torque and proprioceptive info is encoded at high-frequency, and actions are also predicted at this high frequency conditioned on the concatenated outputs of each module. They show experiments in all three areas, with baselines involving changing the input modalities and changing the frequencies of each. There are some real world experiments that ablate only the spatial resolutions.

**Summary Of Recommendation:**

The three separate claims in the paper could each use a bit more attention, especially the temporal resolution and VLM generalization points. It would also be good if the authors could use a more diverse range of dynamic tasks to test out the benefits of the high frequency inputs, and to try recurrent policy architectures since these are more commonly used in practice.

---

> ### Author Response · Authors · 2023-08-15
> **Can we help address any other concerns?**
>
> Dear Reviewer,
>
> Thank you for your detailed and thoughtful review. Since the paper discussion phase ends soon, we would like to check if our response has addressed your concerns. Please let us know if there are some other questions we can resolve.
>
> Thanks,
>
> Paper Authors

---

### Official Review · Reviewer_7F3D · 2023-07-21

**Confidence:** 5
**Originality:** Good
**Technical Quality:** Excellent
**Clarity Of Presentation:** Excellent
**Impact:** 4

**Recommendation:**

Strong Accept: I recommend accepting the paper and will argue for my recommendation even if other reviewers hold a different opinion.

**Review:**

This paper is extremely well motivated, has crisp hypotheses, a simple implementation, and clean experiments that validate every part of the proposed approach. The RLBench and CMU Ballbot results in simulation are convincing, and are a great way for future work to build on this approach. The real-world experiments while a bit toy (peg insertion, block manipulation) highlight the precise/coarse manipulation distinction, and validate that the timing/multi-resolution nature of the approach holds up in the real-world.

The ablations on module capacity, effect of pretraining, fusion mechanisms, and data augmentation all work to further complement this paper.

---

The one concern I have with this work is around the extent of the engineering/push to get temporal resolution experiments running at high frequency. At the scale of the models used (< 300M parameters), I would expect an ability to hit at least 50 Hz control on a 1080Ti with modern CPU compute; if you could run the entire policy at 50 Hz/75 Hz, then the gains seen in this work would probably go away, right?

---
EDIT (Post-Rebuttal): In light of the new experiments and comparisons, I think this is very strong work, and am raising my score to reflect that!

**Quality Of The Limitations Section:**

Limitations are addressed clearly

**Questions For Rebuttal:**

Can you provide more details about the implementation / real-robot stack? How are you reading/parallelizing I/O inputs from the various modalities, and how are you processing inputs through each of the different neural modules? What’s the delta in ceiling control frequency if we were to move up to even one generation later GPU (a 2080 Ti)?

**Robotics Focus:**

Sufficient demonstration on hardware

**Summary Of Paper:**

This strong paper introduces a framework for learning real-time language-conditioned visuomotor manipulation policies that balance global reasoning with high frequency reactive control. The key insight of this work is simple — different input modalities shape the frequency of control actions in different ways. VLMs provide generalization and global reasoning, but are slow. Features from wrist cameras are less necessary for broad task execution, but require higher-frequency handling to compensate for changes in immediate visual surroundings. Finally, proprioceptive + force-torque features dictate the most fine-grained behaviors during execution, and require higher frequency (but generally “lower capacity”) handling.

Results on precise and coarse manipulation tasks in simulation and the real-world validate the necessity of the multi-resolution handling compared to ablations and state of the art BC algorithms.

**Summary Of Recommendation:**

This is a clean and strong paper. I advocate for acceptance at this time, with a plan to push for strong acceptance after resolving the above questions!

---

### Official Review · Reviewer_pRJN · 2023-07-22

**Confidence:** 4
**Originality:** Very Good
**Technical Quality:** Very Good
**Clarity Of Presentation:** Very Good
**Impact:** 4

**Recommendation:**

Weak Accept: I recommend accepting the paper, but will not argue for my recommendation if the majority of other reviewers have a different opinion.

**Review:**

The framework proposed by the authors is quite novel and addresses a very important and practical problem, multi-resolution sensing in real robot learning. The empirical results in the work are very extensive and clearly show the advantages of the proposed method. The authors also included rigorous ablation studies to validate the importance of each design choice.

I have one concern regarding the work. I think the authors should also include comparisons to prior works like RT-1 and BC-Z in the real-world evaluations since RT-1 and BC-Z are designed to be real robot learning algorithms. Including such comparisons would make the paper more convincing.1

**Quality Of The Limitations Section:**

Limitations are addressed clearly

**Questions For Rebuttal:**

1. Compare the method to RT-1 and BC-Z in the real world experiments.

**Robotics Focus:**

Sufficient demonstration on hardware

**Summary Of Paper:**

The paper proposes a framework for learning generalizable language-conditioned multi-task policies that utilize sensing at different spatial and temporal resolutions. The proposed framework consists of two main components:

1. A multi-modal sensory fusion module that combines information from multiple sensing modalities at different spatial and temporal resolutions.

2. A multi-task policy learning module that learns to control the robot to perform a variety of tasks using the fused sensory information.


The authors conducted extensive experiments in three domains of manipulation tasks: coarse, precise, and dynamic. They showed that their approach significantly outperforms recent multi-task baselines.
The authors also showed that their approach generalizes well to visual and geometric variations in target objects and to varying interaction forces.

**Summary Of Recommendation:**

Based on my comments above, I think the paper is of high quality despite lacking some additional comparison. Therefore, I vote for a weak accept.

Post-rebuttal update:

I have read the author response and appreciate the additional experiments. I would like to keep my rating as weak accept.

---

> ### Author Response · Authors · 2023-08-15
> **Can we help address any other questions?**
>
> Dear Reviewer pRJN,
>
> Thank you for your thoughtful review. We have added the additional baselines as you suggested. Please let us know if there are any other questions we can help resolve.

---

### Decision · Program_Chairs · 2023-08-30

**Decision:**

Accept (Poster)

**Comment:**

This submission received mixed reviews.  After the rebuttal, most reviewers became more positive, though reviewer VHHQ did not participate in the discussion.  The AC has read the paper, reviews, and rebuttals, and found that reviewer VHHQ's concerns are reasonably addressed by the rebuttal.  Therefore, the AC agrees with the majority of reviewers and recommends acceptance.  The authors are encouraged to further revise the paper based on the reviews in the camera-ready version.